# Efficacy and Safety of 5-Aminolevulinic Acid Combined with Iron on Skeletal Muscle Mass Index and Physical Performance of Patients with Sarcopenia: A Multicenter, Double-Blinded, Randomized-Controlled Trial (ALADDIN Study)

**DOI:** 10.3390/nu15132866

**Published:** 2023-06-24

**Authors:** Yoshifumi Tamura, Hideyoshi Kaga, Yasuko Abe, Hidenori Yoshii, Hiroaki Seino, Toru Hiyoshi, Nobuichi Kuribayashi, Ikuo Inoue, Hirotaka Watada

**Affiliations:** 1Department of Sports Medicine and Sportology, Graduate School of Medicine, Juntendo University, 2-1-1 Hongo, Bunkyo-ku, Tokyo 113-8421, Japan; 2Sportology Center, Juntendo University Graduate School of Medicine, Tokyo 113-8421, Japan; 3Department of Metabolism and Endocrinology, Graduate School of Medicine, Juntendo University, 2-1-1 Hongo, Bunkyo-ku, Tokyo 113-8421, Japan; 4Faculty of International Liberal Arts, Juntendo University, 2-1-1 Hongo, Bunkyo-ku, Tokyo 113-8421, Japan; 5Department of Internal Medicine, Yokufukai Hospital, Social Welfare Foundation, Tokyo 168-0071, Japan; 6Department of Insured Medical Care Management, Graduate School of Medical and Dental Sciences, Tokyo Medical and Dental University, Tokyo 113-8519, Japan; 7Department of Medicine, Diabetology and Endocrinology, Juntendo Tokyo Koto Geriatric Medical Center, Tokyo 136-0075, Japan; 8Seino Internal Medical Clinic, Fukushima 963-8851, Japan; 9Division of Diabetes and Endocrinology, Japanese Red Cross Medical Center, Tokyo 150-893, Japan; 10Misaki Naika Clinic, Chiba 274-0805, Japan; 11Preventive Medicine Research Center, Saitama Medical University Hospital, Saitama 350-0495, Japan; 12Department of Endocrinology and Diabetology, Saitama Medical University School of Medicine, Saitama 350-0495, Japan

**Keywords:** 5-aminolevulinic acid, hand grip, iron, sarcopenia, skeletal muscle mass index, sodium ferrous citrate

## Abstract

Sarcopenia is a geriatric syndrome characterized by decreased physical performance, muscle mass, and strength. Since the intake of 5-aminolevulinic acid (ALA) with iron can increase muscle mass and mitochondria in mice and elevate physical exercise performance in humans, the beneficial effects of ALA in patients with sarcopenia are expected, but this remains unexplored in the literature. This study aimed to assess the efficacy and dose dependency of ALA combined with iron in sarcopenia by measuring skeletal muscle mass index (SMI). Subjects with sarcopenia were enrolled and randomized into the ALA and sodium ferrous citrate (SFC) intake groups (ALA50/SFC29, ALA100/SFC29, ALA150/SFC29, ALA 100/SFC57, and ALA0/SFC29 placebo) and ingested the assigned study food for 12 weeks. The primary endpoint, the change in SMI from baseline to week 12, did not differ significantly between the groups. Hand grip significantly increased or tended to increase from baseline after 12 weeks with all doses of ALA or SFC compared with the placebo group. No consistent changes were observed in the other endpoints, including calf circumference, physical function, or quality of life (QOL). Although this study suggests safe administration and the possibility of ALA improving hand grip strength in patients with sarcopenia, further investigation is required.

## 1. Introduction

Sarcopenia is a geriatric syndrome characterized by decreased physical performance and a decline in overall muscle mass and strength, mainly due to aging; Rosenberg first proposed it in 1989 [1,2]. The definition and diagnostic criteria for sarcopenia are continuously updated. Sarcopenia was first defined as a significant decline in appendicular skeletal muscle mass. However, since a decline in muscle strength or physical performance was more strongly associated with sarcopenia-related outcomes such as a decrease in activities of daily living (ADL), falls, hospitalization, and death rather than the decline of skeletal muscle mass, the European Working Group on Sarcopenia in Older People defined sarcopenia as a syndrome characterized by progressive and generalized loss of skeletal muscle mass and strength with a risk of adverse outcomes such as physical disability, poor quality of life (QOL), and death, and proposed diagnostic criteria based on walking speed, hand grip, and muscle mass [3]. In Asia, the Asian Working Group for Sarcopenia (AWGS) proposed the diagnostic criteria AWGS 2014 [4] and AWGS 2019 [5], considering ethical differences in body constitution and physical function.

To prevent sarcopenia, appropriate nutrient intake, especially protein consumption (1.0 g per kg ideal body weight per day or more), is considered necessary [6,7]. Nutritional intervention with essential amino acids or pharmacotherapy with selective androgen receptor modulators is mainly conducted to treat sarcopenia; however, evidence supporting these treatments is still insufficient.

A natural amino acid synthesized in animals and plants, 5-aminolevulinic acid (ALA), is a porphyrin precursor. Porphyrins are synthesized by linking eight ALA molecules. Porphyrins, which bind to iron ions, are known as heme. Heme acts as a cofactor for many proteins, such as hemoglobin and cytochrome, and plays essential roles in energy metabolism in the mitochondria. It was reported that the intake of ALA with iron increased mouse muscle mass and mitochondria and elevated physical exercise performance in humans [8,9]. Based on this, a beneficial effect of ALA for patients with sarcopenia was expected; however, there is insufficient evidence supporting this information in the literature. Since ALA was considered to improve sarcopenia through conjugation of eight ALA molecules and ferrous into heme [8], this study aimed to assess the efficacy and dose dependency of ALA combined with iron in patients diagnosed with sarcopenia by measuring the skeletal muscle mass index (SMI). As muscle strength assessed by hand grip was associated with physical and mental health-related QOL in elderly [10], this study also assessed the hand grip and other physical performance as secondary endpoints.

## 2. Materials and Methods

### 2.1. Study Design

This was a multicenter, double-blinded, randomized-controlled trial conducted at seven medical institutions (Appendix A) in Japan. The study and its protocols were inspected and approved in December 2020 by the Juntendo University Certified Review Board (J20-019), which obtained certification from the Minister of Health, Labor, and Welfare in Japan. This study was registered in the Japan Registry of Clinical Trials (jRCT) (registration number: jRCTs031200433) in March 2021 after approval from the certified review board according to the requirements of the Clinical Trials Act. Subject enrollment was conducted between March 2021, after registration in the jRCT, and March 2022. Each enrolled subject was followed up for 12 weeks. The study was conducted in accordance with the Declaration of Helsinki, the Clinical Trials Act, and other current legal regulations in Japan. Written informed consent was obtained from all enrolled patients who met the eligibility criteria before treatment. To avoid bias and ensure quality, data collection, management, monitoring, auditing, and statistical analyses were performed by a third-party entity (EviPRO Co., Ltd., Tokyo, Japan). All authors had access to the study data and reviewed and approved the final manuscript.

### 2.2. Subject Population

Subjects who met all of the inclusion criteria in this study, which reflected the diagnostic criteria for sarcopenia, were included. The main inclusion criteria were as follows: (1) male and female participants aged 65 years or older who provided consent, and (2) male and female participants whose walking speed was less than 1 m/sec, or men with a hand grip of less than 25 kg or women with a hand grip of less than 20 kg at the time of providing consent, and (3) body mass index (BMI) less than 18.5 kg/m^2^, men whose lower leg circumference was less than 34 cm or women whose lower leg circumference was less than 33 cm at the time of consent, and (4) men whose SMI measured by bioimpedance analysis (BIA) method was less than 7.0 kg/m^2^, or women whose SMI measured by BIA method was less than 5.7 kg/m^2^ at the time of giving consent. SMI was measured using an InBody270 (InBody Japan Inc., Tokyo, Japan). Details of the inclusion and exclusion criteria are provided in Appendix A.

### 2.3. Randomization and Study Intervention

After obtaining informed consent, eligible subjects were randomly assigned to one of the following five groups at approximately a 1:1:1:1:1 ratio: 50 mg ALA phosphate and 29 mg sodium ferrous citrate (SFC) food intake group (ALA 50/SFC 29 group), 100 mg ALA phosphate and 29 mg SFC food intake group (ALA 100/SFC 29 group), 150 mg ALA phosphate and 29 mg SFC food intake group (ALA 150/SFC 29 group), 100 mg ALA phosphate and 57 mg SFC food intake group (ALA 100/SFC 57 group), and 0 mg ALA phosphate and 29 mg SFC food intake group (placebo group). All study foods included 50 mg zinc yeast (5 mg zinc), since ALA was considered to improve sarcopenia through conjugation of eight ALA molecules and ferrous into heme [8], and zinc was required for enzymatic activity of porphobilinogen synthase that acted in a step of ALA condensation [11]. Randomization was performed using a computer-based dynamic allocation method with a minimization procedure to balance one allocation factor (sex) across groups. All study foods were sealed in visually indistinguishable capsules and provided to the subjects. The subjects were asked to consume the assigned study food for 12 weeks and were followed up for this duration, with observation points at baseline and weeks four, eight, and twelve. Detailed observation schedules and items are listed in Appendix A.

### 2.4. Study Outcomes

The primary endpoint was the change in SMI measured using BIA from baseline to week 12. SMI was measured using an InBody270 (InBody Japan Inc., Tokyo, Japan). Secondary endpoints included body weight, BMI, basal metabolic rate (BMR), hand grip strength, lower leg circumference, physical activity, 6 m walking distance (6 MWD), 5-repetition sit-to-stand test (5STS), Short Physical Performance Battery (SPPB) [12], Medical Outcomes Study Short-Form 36-Item Health, Survey (SF-36) [13], Beck Depression Inventory (BDI) [14], and EuroQol 5 Dimension-5 Level (EQ-5D-5L) [15]. The SPPB is an objective assessment tool for evaluating lower-extremity function in older individuals by measuring balancing, walking, and sit-to-stand functions. The SF-36 is a questionnaire on health-related QOL consisting of 36 items and eight health domains: physical functioning (ten items), bodily pain (two items), role limitations due to physical health problems (four items), role limitations due to personal or emotional problems (four items), emotional well-being (five items), social functioning (two items), energy/fatigue (four items), and general health perceptions (five items). The scores for each domain range from 0 to 100, with a higher score indicating a more favorable health state. The BDI questionnaire assesses the severity of depressive symptoms; it consists of 21 items. The EQ-5D-5L assesses health-related QOL and consists of five items. The full list of endpoints is provided in Appendix A. All tests at week 0 (baseline) were conducted before the initiation of the study food intake.

### 2.5. Sample Size Calculation and Statistical Analysis

The target sample size in this study was determined based on a previous cross-sectional study investigating the prevalence of sarcopenia in an elderly Japanese population aged 65 years or older with type 2 diabetes mellitus [16]. The study reported that SMI was 6.35 kg/m^2^ for males and 5.32 kg/m^2^ for females in patients with sarcopenia and 7.57 kg/m^2^ for males and 6.47 kg/m^2^ for females in the healthy population; therefore, it was 1.22 kg/m^2^ lower for males and 1.15 kg/m^2^ lower for females in patients with sarcopenia compared with the healthy population. From these data, we assumed that the intergroup difference in change in SMI between the placebo group and the ALA food intake group was 1.15 kg/m^2^. The standard deviation (SD) to a shift in SMI was assumed to be 0.78 kg/m^2^, the largest SD in the cross-sectional study (SD in healthy females). Under these assumptions, 14 participants per group would provide a power of 90% to detect intergroup differences using Dunnett’s test at a 5% significance level. Considering the nature of this study and dropout or discontinuation during the observation period, 20 subjects per group were determined as the target sample size, yielding a total sample size of 100.

Analyses of the primary and secondary endpoints were performed with data from the full analysis set population, including all subjects enrolled in this study who were subsequently randomized to one of the study groups but excluding those with a significant study protocol violation (e.g., registration without consent or registration out of the enrollment period). Sensitivity analysis of the primary endpoint was performed with the per-protocol set by excluding subjects with a protocol violation, such as violation of eligibility criteria, use of prohibited drugs, or poor medication adherence to the study or control agents (<70% or >120%). The safety analysis included all subjects registered in this study who adhered to the study’s food intake. All tests were two-sided, and statistical significance was set at *p* < 0.05. The primary endpoint, change in SMI from baseline to week 12, was tested by analysis of covariance, with the treatment groups as the fixed effect and the allocation factor (sex) as a covariate. Multiplicity was adjusted using a sequential Dunnett’s test, with the placebo group as a control. For the subjects’ backgrounds, comparisons between the five groups were performed using analysis of variance for continuous variables or the chi-square test for categorical variables. For the analysis of physical examinations and functions among the secondary endpoints, the adjusted mean change of each continuous variable was estimated using models for repeated measures (MMRM) with an unstructured covariance structure with treatment group, time, the interaction between treatment group and time, and allocation factor as fixed effects and enrolled subjects as random effects. If the results of the MMRM using the unstructured covariance structure failed to converge, Toeplits, autoregressive, or compound symmetry covariance structures were sequentially used. For the analysis of the participants’ QOL among the secondary endpoints, a two-sample *t*-test was performed for intergroup comparisons with the placebo group. Fisher’s exact test was performed for intergroup comparisons with the placebo group to analyze adverse events. All statistical analyses were performed by a third-party entity (EviPRO Co., Ltd.) using SAS version 9.4 (SAS Institute Inc., Cary, NC, USA), according to the statistical analysis plan that had been developed in a blinded manner before the database lock.

## 3. Results

### 3.1. Baseline Characteristics of Subjects

A total of 277 candidates were screened, after which 177 were excluded from the study. Of the 177 excluded subjects, 129 did not meet the eligibility criteria and 48 did not provide consent to participate in this study. Finally, 100 participants were randomly assigned to the intervention groups. As a result, 20 subjects were assigned to the placebo group, 18 to the ALA 50/SFC 29 group, 22 to the ALA 100/SFC 29 group, 19 to ALA 150/SFC 29 group, and 21 to ALA 100/SFC 57 group (Figure 1). The baseline characteristics of the registered participants were well-balanced among the groups (Table 1).

### 3.2. Change in Skeletal Muscle Mass Index

The primary endpoint in this study, the change in SMI from baseline to week 12, did not differ significantly between the groups (Figure 2). It was significantly higher in the ALA 100/SFC 29 group than in the placebo group at week eight (*p* = 0.040).

### 3.3. Change in Physical Examination

No significant intergroup differences were observed in changes in body weight and BMI from baseline to week 12, while calf circumference significantly increased from baseline to week 12 in the ALA 100/SFC 29 group compared with the placebo group (Table 2).

### 3.4. Change in Physical Function

Hand grip significantly increased or tended to increase from baseline at week 12 with all doses of ALA or SFC compared with the placebo group (Table 3). Decrease of physical activity from baseline to week 12 was significantly suppressed in the ALA 50/SFC 29 group and ALA 100/SFC 29 group, and tended to be suppressed in the ALA 100/SFC 57 group, compared with the placebo group. The 5STS significantly decreased or tended to decrease from baseline to week four in the ALA 50/SFC 29 group and ALA 100/SFC 29 group compared with the placebo group; however, no significant intergroup difference in the change in 5STS was observed at week 12. No significant intergroup differences were observed in the changes in the BMR, 6 MWD, and SPPB. 

### 3.5. Change in Subjects’ Quality of Life

No significant intergroup differences were observed in the changes in the SF-36 and BDI scores (Table 4). The visual analog scale score in the EQ-5D-5L significantly increased from baseline to week 12 in the ALA 100/SFC 29 group compared to the placebo group.

### 3.6. Safety

The adverse events that occurred during the study are listed in Table 5. No deaths were reported in any group, and the frequency of non-serious or serious adverse events did not differ between the groups.

## 4. Discussion

This study assessed the efficacy and dose dependence of ALA in sarcopenia. No significant intergroup difference in the change in SMI compared to the placebo group was observed at the 12-week follow-up; this may have been due to the small sample size (20 cases per group), short intervention period (12 weeks), and large individual differences in the change in SMI, resulting in a large SD. In addition, the target sample size in this study was determined based on a previous cross-sectional study that investigated the prevalence of sarcopenia among the elderly Japanese population [16]. In accordance with the previous cross-sectional study, change and SD in SMI were assumed as 1.15 kg/m^2^ and 0.78 kg/m^2^, respectively. However, the actual change in SMI from baseline to week twelve was ≤ 1/10 (at maximum 0.10 ± 0.06 kg/m^2^ in the ALA 100/SFC 29 group at week eight) than the assumption; this may have resulted in insufficient power. Further investigation is required in the future with a more appropriate sample size and extended intervention period based on the calculation of the change and SD in this study.

Among the secondary endpoints, hand grip significantly increased or tended to increase from baseline to week 12 with all doses of ALA or SFC compared with the placebo group. Although it was reported that hand grip was associated with physical and mental health-related QOL in elderly [10], there were no consistent changes in other endpoints, including calf circumference, 5STS, and QOLs in this study; this may also be due to the relatively small sample size and short intervention period. The obvious beneficial effects of ALA on these endpoints were not demonstrated in this study and require further investigation.

No safety concerns were observed for any doses of ALA or SFC. A previous study reported that a dose increase of ALA from 100 mg to 200 mg did not cause serious adverse events or increase the frequency of non-serious adverse events compared with the placebo [17]. In addition, excess iron intake causes gastrointestinal symptoms, such as nausea, vomiting, hepatic dysfunction, and rashes [17]. The upper limit of iron intake was determined as 50–55 mg/day for adult males and 40–45 mg/day for adult females in the dietary reference intake for Japanese issued by the Ministry of Health, Labor and Welfare in Japan [7], and the mean iron intake for adults in Japan was reported as 7.8 mg/day in a nationwide survey in 2017 [18]. Therefore, the 29 mg or 57 mg per day intake of SFC (iron 3.0 mg or 6.0 mg per day) in this study was not considered excessive. The results of this study suggest that an increase in the dose of ALA or SFC did not increase the incidence of adverse events, which is consistent with previous evidence, and this study confirmed the safety of ALA and SFC.

This study has several limitations. First, as described above, owing to the small sample size and a minor change in the primary endpoint than assumed in the sample size calculation, the power to detect intergroup differences might be insufficient. Further investigation is required in the future with a more appropriate sample size and extended intervention period based on the calculation of the change and SD in this study. Second, SMI was measured using the BIA method in this study. Although measurement with a more accurate method (dual-energy X-ray absorptiometry) might be more appropriate to assess the effect of ALA on sarcopenia under conditions close to actual medical care, this study employed the BIA method. This study used InBody270, whose accuracy is relatively high compared to that of the BIA method; however, the employment of the BIA method might be a reason for the large SD. Third, no consistent change in the endpoints, except for hand grip, was observed. Although hand grip strength significantly increased or tended to increase from baseline to week 12 with all doses of ALA or SFC compared to the placebo group, no obvious dose dependency was observed. However, the effects of certain external factors cannot be ignored. Since this study was conducted during the coronavirus pandemic of 2019, physical activity or dietary habits might have been affected by the pandemic, and these changes might have affected the results. Fourth, the effects of other medications and complications were not assessed in this study. Since polypharmacy is an Important problem in elderly [19,20,21], especially in case of multimorbidity [19,22], further investigation is required to establish treatment strategy for sarcopenia to improve multiple aspects of physical composition and functions to reduce the number of complications and medications. Fifth, this study was conducted in Japan’s medical institutions; all registered participants were Japanese. Therefore, the generalizability of the results to other countries or ethnicities should be carefully considered.

## 5. Conclusions

The intake of ALA combined with SFC did not significantly increase the SMI but suggested the possibility of improving hand grip with safety profile in patients with sarcopenia. Further investigations with a more appropriate sample size and extended intervention period are required.

## Figures and Tables

**Figure 1 nutrients-15-02866-f001:**
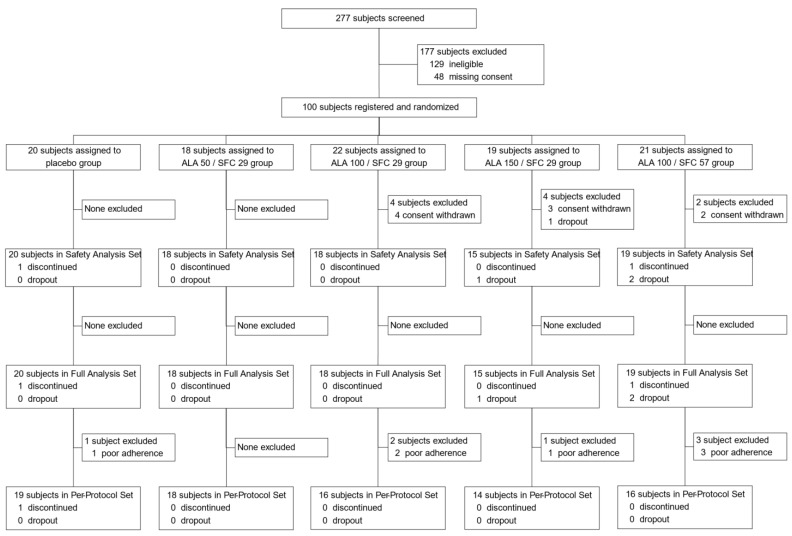
Study flowchart showing subject enrollment, allocation, and analysis. ALA, 5-aminolevulinic acid; SFC, sodium ferrous citrate.

**Figure 2 nutrients-15-02866-f002:**
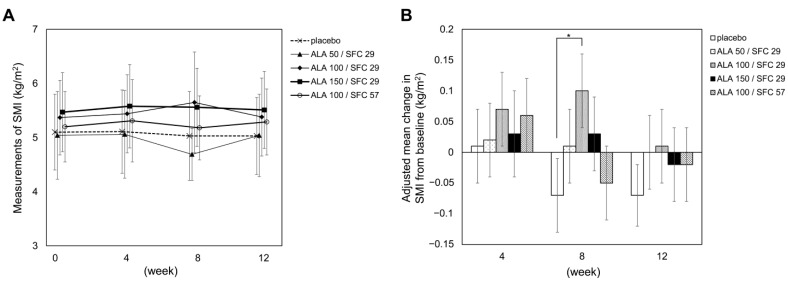
Change in skeletal muscle mass index (SMI). Measurement of SMI (**A**) and adjusted mean
change in SMI (**B**) are shown. A two-sample *t*-test was performed for pairwise intergroup comparisons of the measurements. Adjusted mean change was estimated using models for repeated measures (MMRM) with an unstructured covariance structure with treatment group, time, the interaction between treatment group and time, and allocation factor as fixed effects, and enrolled subjects as random effects. If the results of the MMRM using the unstructured covariance structure failed to converge, Toeplits, autoregressive, or compound symmetry covariance structures were sequentially used. * represents *p* < 0.05. ALA, 5-aminolevulinic acid; SFC, sodium ferrous citrate; SMI, skeletal muscle mass index.

**Table 1 nutrients-15-02866-t001:** Backgrounds of subjects.

	Placebo Group	ALA 50/SFC 29 Group	ALA 100/SFC 29 Group	ALA 150/SFC 29 Group	ALA 100/SFC 57 Group	*p*-Value
Age (years)	82.1 ± 6.1 (20)	80.1 ± 6.7 (18)	80.3 ± 8.3 (18)	77.7 ± 6.7 (15)	81.8 ± 6.5 (19)	0.36
Female sex	16 (80.0)	14 (77.8)	11 (61.1)	10 (66.7)	13 (68.4)	0.69
Height (cm)	150.9 ± 6.9 (20)	153.7 ± 6.8 (18)	156.5 ± 7.6 (18)	155.0 ± 7.4 (15)	150.9 ± 7.8 (19)	0.08
Body weight (kg)	42.4 ± 5.7 (20)	42.3 ± 6.0 (18)	46.8 ± 7.2 (18)	46.4 ± 6.4 (15)	43.7 ± 5.8 (19)	0.08
Calf circumference (cm) *	29.5 ± 1.7 (20)	29.6 ± 2.1 (18)	29.8 ± 1.4 (18)	30.4 ± 1.6 (15)	29.7 ± 1.9 (19)	0.59
SMI (kg/m^2^)	5.10 ± 0.70 (20)	5.04 ± 0.81 (18)	5.37 ± 0.69 (18)	5.47 ± 0.73 (15)	5.20 ± 0.65 (19)	0.37
Hand grip (kg) **	17.8 ± 4.5 (20)	18.5 ± 6.0 (18)	19.9 ± 2.8 (18)	19.1 ± 4.1 (15)	16.7 ± 3.9 (19)	0.23
5STS (sec)	13.5 ± 5.5 (20)	16.1 ± 8.8 (17)	13.4 ± 4.9 (17)	12.7 ± 3.9 (13)	16.6 ± 7.2 (18)	0.29
6 MWD (sec)	11.8 ± 3.1 (3)	14.3 ± 4.9 (3)	6.2 ± 0.5 (2)	7.4 ± 1.2 (2)	11.3 ± 4.6 (4)	0.21
SPPB score	9.7 ± 2.1 (20)	9.1 ± 3.1 (18)	9.3 ± 2.5 (18)	9.1 ± 2.8 (15)	8.7 ± 2.6 (19)	0.84

Data are presented as mean ± standard deviation (n) or the number of subjects (%). *p*-values for comparison between the five groups by analysis of variance for continuous variables or the chi-squared test for categorical variables are presented. * Calf circumference was calculated as the mean of the left and right maximum calf circumference. ** Hand grip was calculated as the mean of left- and right-hand grips. ALA, 5-aminolevulinic acid; SFC, sodium ferrous citrate; SMI, skeletal muscle mass index; 6 MWD, 6 min walking distance; 5STS, 5-repetition sit-to-stand test; SPPB, Short Physical Performance Battery.

**Table 2 nutrients-15-02866-t002:** Change in physical examination.

	Week		Placebo Group	ALA 50/SFC 29 Group	ALA 100/SFC 29 Group	ALA 150/SFC 29 Group	ALA 100/SFC 57 Group
Body weight (kg)	0	measurement	42.4 ± 5.7 (20)	42.3 ± 6.0 (18)	46.8 ± 7.2 (18)	46.4 ± 6.4 (15)	43.7 ± 5.8 (19)
	4	measurement	42.6 ± 5.7 (20)	42.5 ± 6.1 (18)	47.2 ± 7.5 (18)	47.0 ± 5.8 (13)	44.0 ± 5.7 (18)
		adjusted mean change from baseline/*p*-value vs. placebo	0.1 (0.2)	0.2 (0.2)/0.87	0.4 (0.2)/0.37	−0.2 (0.3)/0.41	0.0 (0.2)/0.70
	8	measurement	42.5 ± 8.1 (8)	40.8 ± 5.5 (7)	48.3 ± 8.7 (8)	47.7 ± 5.4 (9)	43.1 ± 6.8 (6)
		adjusted mean change from baseline/*p*-value vs. placebo	0.0 (0.2)	0.0 (0.2)/0.97	0.2 (0.2)/0.43	−0.2 (0.2)/0.57	0.0 (0.2)/0.94
	12	measurement	42.4 ± 5.8 (20)	42.4 ± 6.0 (18)	46.8 ± 7.1 (18)	46.9 ± 5.5 (14)	44.2 ± 5.4 (17)
		adjusted mean change from baseline/*p*-value vs. placebo	−0.1 (0.3)	0.1 (0.3)/0.67	0.0 (0.3)/0.87	−0.4 (0.3)/0.45	−0.3 (0.3)/0.53
BMI (kg/m^2^)	0	measurement	18.6 ± 2.2 (20)	17.9 ± 2.1 (18)	19.1 ± 2.3 (18)	19.3 ± 2.3 (15)	19.2 ± 2.2 (19)
	4	measurement	18.7 ± 2.2 (20)	18.0 ± 2.1 (18)	19.2 ± 2.2 (18)	19.5 ± 1.9 (13)	19.2 ± 2.3 (18)
		adjusted mean change from baseline/*p*-value vs. placebo	0.1 (0.1)	0.1 (0.1)/0.97	0.1 (0.1)/0.62	−0.1 (0.1)/0.31	0.0 (0.1)/0.59
	8	measurement	18.7 ± 2.7 (8)	18.0 ± 2.1 (7)	19.2 ± 2.3 (8)	19.7 ± 1.6 (9)	18.8 ± 3.5 (6)
		adjusted mean change from baseline/*p*-value vs. placebo	0.0 (0.1)	0.0 (0.1)/0.80	0.1 (0.1)/0.61	−0.1 (0.1)/0.49	0.0 (0.1)/0.91
	12	measurement	18.6 ± 2.1 (20)	17.9 ± 2.1 (18)	19.1 ± 2.2 (18)	19.5 ± 1.8 (14)	19.2 ± 2.4 (17)
		adjusted mean change from baseline/*p*-value vs. placebo	0.0 (0.1)	0.0 (0.1)/0.71	0.0 (0.1)/0.98	−0.2 (0.1)/0.39	−0.1 (0.1)/0.51
Calf circumference (cm) *	0	measurement	29.5 ± 1.7 (20)	29.6 ± 2.1 (18)	29.8 ± 1.4 (18)	30.4 ± 1.6 (15)	29.7 ± 1.9 (19)
	4	measurement	29.6 ± 2.0 (20)	29.8 ± 1.9 (18)	30.4 ± 2.0 (18)	30.8 ± 1.4 (14)	29.8 ± 1.8 (18)
		adjusted mean change from baseline/*p*-value vs. placebo	0.0 (0.2)	0.0 (0.3)/0.99	0.5 (0.2)/0.15	0.1 (0.3)/0.84	−0.1 (0.2)/0.60
	8	measurement	- (0)	- (0)	32.3 (1)	30.5 ± 0.4 (2)	- (0)
		adjusted mean change from baseline/*p*-value vs. placebo	-	-/-	1.4 (0.9)/-	0.1 (0.6)/-	-/-
	12	measurement	29.3 ± 2.2 (20)	29.7 ± 2.4 (18)	30.4 ± 1.8 (18)	30.7 ± 1.5 (14)	29.8 ± 1.9 (16)
		adjusted mean change from baseline/*p*-value vs. placebo	−0.2 (0.2)	0.0 (0.3)/0.49	0.5 (0.2)/0.027	0.0 (0.3)/0.48	−0.3 (0.3)/0.87

Data are presented as mean ± standard deviation (n) for measurements and mean (standard error) for adjusted mean change from baseline. The adjusted mean change of each continuous variable was estimated using models for repeated measures (MMRM) with an unstructured covariance structure with treatment group, time, the interaction between treatment group and time, and allocation factor as fixed effects and enrolled subjects as random effects. If the results of the MMRM using the unstructured covariance structure failed to converge, Toeplits, autoregressive, or compound symmetry covariance structures were sequentially used. * Calf circumference was calculated as the mean of the left and right maximum calf circumference. ALA, 5-aminolevulinic acid; SFC, sodium ferrous citrate; BMI, body mass index.

**Table 3 nutrients-15-02866-t003:** Change in physical function.

	Week		Placebo Group	ALA 50/SFC 29 Group	ALA 100/SFC 29 Group	ALA 150/SFC 29 Group	ALA 100/SFC 57 Group
BMR (kcal)	0	measurement	1080.1 ± 100.1 (20)	1087.5 ± 104.6 (18)	1139.1 ± 101.2 (18)	1147.6 ± 129.5 (15)	1100.4 ± 85.8 (19)
	4	measurement	1086.1 ± 104.6 (20)	1093.5 ± 105.9 (18)	1144.1 ± 104.1 (18)	1160.7 ± 124.9 (13)	1112.3 ± 94.0 (18)
		adjusted mean change from baseline/*p*-value vs. placebo	5.8 (6.7)	5.9 (7.0)/0.99	4.9 (6.8)/0.93	−1.8 (7.9)/0.46	4.0 (6.9)/0.85
	8	measurement	1074.1 ± 113.2 (8)	1045.4 ± 80.8 (7)	1173.0 ± 122.1 (8)	1168.0 ± 119.3 (9)	1117.8 ± 80.7 (6)
		adjusted mean change from baseline/*p*-value vs. placebo	−2.4 (7.1)	1.8 (7.5)/0.67	10.7 (6.9)/0.19	2.8 (6.8)/0.59	0.8 (7.8)/0.75
	12	measurement	1077.6 ± 102.1 (20)	1091.9 ± 104.3 (18)	1139.7 ± 98.5 (18)	1153.6 ± 123.5 (14)	1110.1 ± 79.5 (17)
		adjusted mean change from baseline/*p*-value vs. placebo	−2.6 (6.4)	4.3 (6.7)/0.44	0.6 (6.5)/0.72	−5.6 (7.4)/0.76	−4.0 (6.7)/0.88
Hand grip (kg) *	0	measurement	17.8 ± 4.5 (20)	18.5 ± 6.0 (18)	19.9 ± 2.8 (18)	19.1 ± 4.1 (15)	16.7 ± 3.9 (19)
	4	measurement	17.9 ± 4.5 (19)	18.7 ± 6.1 (18)	20.1 ± 3.5 (18)	19.9 ± 4.6 (14)	16.5 ± 3.6 (18)
		adjusted mean change from baseline/*p*-value vs. placebo	0.0 (0.4)	0.3 (0.4)/0.53	0.3 (0.4)/0.50	0.9 (0.4)/0.07	−0.4 (0.4)/0.50
	8	measurement	- (0)	- (0)	21.0 ± 4.3 (2)	20.1 ± 6.0 (2)	- (0)
		adjusted mean change from baseline/*p*-value vs. placebo	-	-/-	1.4 (1.0)/-	1.1 (1.0)/-	-/-
	12	measurement	16.8 ± 4.1 (20)	18.5 ± 6.0 (18)	20.8 ± 4.0 (18)	19.4 ± 4.3 (13)	17.6 ± 3.8 (16)
		adjusted mean change from baseline/*p*-value vs. placebo	−0.8 (0.3)	0.1 (0.4)/0.05	1.0 (0.4)/<0.001	0.2 (0.4)/0.06	0.6 (0.4)/0.005
Physical activity (Ex/day)	0	measurement	20.7 ± 4.8 (18)	19.5 ± 5.5 (17)	19.9 ± 7.8 (15)	19.8 ± 5.2 (12)	15.8 ± 7.4 (19)
	4	measurement	19.4 ± 5.2 (20)	19.8 ± 5.5 (18)	20.6 ± 6.7 (15)	19.6 ± 6.6 (14)	15.8 ± 7.1 (18)
		adjusted mean change from baseline/*p*-value vs. placebo	−0.8 (0.6)	−0.2 (0.6)/0.43	−0.3 (0.6)/0.54	−0.9 (0.7)/0.92	−0.6 (0.6)/0.75
	8	measurement	18.5 ± 5.5 (18)	19.0 ± 6.2 (18)	19.5 ± 7.1 (16)	18.6 ± 6.3 (14)	16.0 ± 6.7 (17)
		adjusted mean change from baseline/*p*-value vs. placebo	−2.3 (0.7)	−1.0 (0.7)/0.17	−2.0 (0.8)/0.73	−1.7 (0.8)/0.58	−0.9 (0.7)/0.16
	12	measurement	17.2 ± 5.5 (18)	18.9 ± 5.9 (18)	19.5 ± 6.6 (15)	17.7 ± 6.8 (13)	15.5 ± 6.1 (16)
		adjusted mean change from baseline/*p*-value vs. placebo	−3.7 (0.7)	−1.0 (0.7)/0.008	−1.5 (0.8)/0.037	−2.7 (0.8)/0.36	−2.0 (0.7)/0.08
5STS (sec)	0	measurement	13.5 ± 5.5 (20)	16.1 ± 8.8 (17)	13.4 ± 4.9 (17)	12.7 ± 3.9 (13)	16.6 ± 7.2 (18)
	4	measurement	14.8 ± 8.6 (20)	14.2 ± 6.3 (17)	12.3 ± 4.3 (18)	12.8 ± 5.1 (13)	15.6 ± 4.9 (17)
		adjusted mean change from baseline/*p*-value vs. placebo	1.5 (1.0)	−1.7 (1.0)/0.022	−1.1 (1.0)/0.06	0.3 (1.2)/0.43	−0.4 (1.0)/0.16
	8	measurement	- (0)	- (0)	- (0)	13.0 ± 2.1 (2)	- (0)
		adjusted mean change from baseline/*p*-value vs. placebo	-	-/-	-/-	−3.5 (4.2)/-	-/-
	12	measurement	13.2 ± 5.5 (20)	13.6 ± 7.4 (17)	12.4 ± 5.3 (18)	12.1 ± 4.2 (14)	13.8 ± 4.3 (16)
		adjusted mean change from baseline/*p*-value vs. placebo	−0.1 (1.0)	−2.2 (1.0)/0.12	−1.2 (1.0)/0.42	−1.0 (1.2)/0.55	−0.8 (1.0)/0.61
6 MWD (sec)	0	measurement	11.8 ± 3.1 (3)	14.3 ± 4.9 (3)	6.2 ± 0.5 (2)	7.4 ± 1.2 (2)	11.3 ± 4.6 (4)
	4	measurement	12.0 ± 6.2 (3)	14.4 ± 5.1 (3)	6.7 ± 0.6 (2)	9.2 ± 0.3 (2)	8.6 ± 0.3 (3)
		adjusted mean change from baseline/*p*-value vs. placebo	0.2 (1.3)	0.1 (1.3)/0.93	0.5 (1.6)/0.91	1.8 (1.6)/0.47	−0.3 (1.3)/0.75
	8	measurement	- (0)	- (0)	- (0)	- (0)	- (0)
		adjusted mean change from baseline/*p*-value vs. placebo	-	-/-	-/-	-/-	-/-
	12	measurement	12.0 ± 5.6 (3)	15.3 ± 9.0 (3)	6.5 ± 1.4 (2)	7.0 ± 0.4 (2)	9.2 ± 0.5 (3)
		adjusted mean change from baseline/*p*-value vs. placebo	0.2 (1.6)	1.0 (1.6)/0.73	0.3 (2.0)/0.98	−0.4 (2.0)/0.84	0.2 (1.6)/0.99
SPPB score	0	measurement	9.7 ± 2.1 (20)	9.1 ± 3.1 (18)	9.3 ± 2.5 (18)	9.1 ± 2.8 (15)	8.7 ± 2.6 (19)
	4	measurement	10.0 ± 2.3 (20)	9.6 ± 2.6 (18)	10.3 ± 1.5 (17)	9.6 ± 2.8 (14)	8.9 ± 2.7 (18)
		adjusted mean change from baseline/*p*-value vs. placebo	0.1 (0.3)	0.4 (0.3)/0.55	0.6 (0.3)/0.30	0.0 (0.4)/0.76	−0.2 (0.3)/0.48
	8	measurement	- (0)	- (0)	- (0)	8.0 ± 1.4 (2)	- (0)
		adjusted mean change from baseline/*p*-value vs. placebo	-	-/-	-/-	3.8 (0.5)/-	-/-
	12	measurement	10.2 ± 1.9 (19)	9.6 ± 3.0 (18)	9.9 ± 2.5 (18)	10.0 ± 2.1 (14)	9.6 ± 2.1 (16)
		adjusted mean change from baseline/*p*-value vs. placebo	0.2 (0.3)	0.4 (0.3)/0.56	0.6 (0.3)/0.38	0.4 (0.4)/0.57	0.1 (0.3)/0.82

Data are presented as mean ± standard deviation (n) for measurements and mean (standard error) for adjusted mean change from baseline. The adjusted mean change of each continuous variable was estimated using models for repeated measures (MMRM) with an unstructured covariance structure with treatment group, time, the interaction between treatment group and time, and allocation factor as fixed effects and enrolled subjects as random effects. If the results of the MMRM using the unstructured covariance structure failed to converge, Toeplits, autoregressive, or compound symmetry covariance structures were sequentially used. * Hand grip strength was calculated as the mean of the left and right grips. ALA, 5-aminolevulinic acid; SFC, sodium ferrous citrate; BMR, basal metabolic rate; 6 MWD, 6 min walking distance; 5STS, 5-repetition sit-to-stand test; SPPB, Short Physical Performance Battery.

**Table 4 nutrients-15-02866-t004:** Change in quality of life.

	Week		Placebo Group	ALA 50/SFC 29 Group	ALA 100/SFC 29 Group	ALA 150/SFC 29 Group	ALA 100/SFC 57 Group
SF-36 (NBS)							
physical functioning	0	Measurement	43.5 ± 11.2 (20)	40.5 ± 15.3 (18)	45.1 ± 8.5 (17)	44.1 ± 11.7 (14)	42.3 ± 13.8 (19)
	12	Measurement	45.8 ± 7.9 (20)	42.4 ± 14.7 (18)	47.0 ± 12.4 (18)	48.9 ± 7.2 (13)	44.1 ± 13.4 (17)
		change from baseline/ *p*-value vs. placebo	2.3 ± 6.3	1.9 ± 6.2/0.84	1.4 ± 8.9/0.73	0.5 ± 6.2/0.43	−1.1 ± 5.1/0.08
role-physical	0	Measurement	42.8 ± 12.8 (20)	38.4 ± 13.7 (17)	46.2 ± 12.3 (17)	41.6 ± 16.9 (14)	44.7 ± 13.3 (19)
	12	Measurement	45.2 ± 10.2 (20)	41.8 ± 14.7 (17)	51.4 ± 10.6 (18)	46.3 ± 14.4 (13)	43.8 ± 13.8 (17)
		change from baseline/ *p*-value vs. placebo	2.3 ± 8.1	4.5 ± 7.8/0.42	4.9 ± 11.7/0.43	−0.2 ± 19.9/0.61	−2.4 ± 13.1/0.19
body pain	0	Measurement	44.9 ± 10.8 (20)	45.1 ± 10.9 (17)	46.3 ± 11.8 (17)	42.6 ± 12.9 (14)	45.8 ± 9.9 (19)
	12	Measurement	46.3 ± 10.3 (20)	48.8 ± 10.2 (17)	50.1 ± 12.6 (18)	47.6 ± 10.6 (13)	49.3 ± 10.5 (17)
		change from baseline/ *p*-value vs. placebo	1.3 ± 10.6	5.3 ± 8.4/0.23	3.9 ± 10.0/0.46	3.2 ± 8.6/0.60	2.1 ± 7.1/0.79
general health perception	0	Measurement	49.1 ± 8.2 (20)	47.9 ± 9.0 (18)	53.7 ± 8.7 (16)	55.8 ± 10.0 (14)	48.9 ± 10.3 (18)
	12	Measurement	50.8 ± 8.3 (20)	50.4 ± 8.6 (18)	57.0 ± 9.1 (18)	58.8 ± 9.3 (12)	51.8 ± 10.7 (17)
		change from baseline/ *p*-value vs. placebo	1.6 ± 7.9	2.5 ± 7.8/0.74	2.6 ± 6.2/0.69	2.7 ± 7.9/0.73	1.7 ± 7.3/0.98
vitality	0	Measurement	49.2 ± 8.8 (20)	49.5 ± 7.1 (18)	53.8 ± 10.5 (17)	50.9 ± 9.4 (14)	50.0 ± 9.5 (18)
	12	Measurement	51.4 ± 9.8 (20)	51.0 ± 8.0 (18)	54.2 ± 9.5 (18)	54.4 ± 13.9 (12)	51.3 ± 9.9 (17)
		change from baseline/ *p*-value vs. placebo	2.1 ± 6.5	1.5 ± 8.6/0.81	−0.4 ± 9.5/0.36	2.2 ± 15.3/0.97	0.4 ± 10.4/0.55
social functioning	0	Measurement	47.6 ± 12.3 (20)	41.1 ± 14.1 (18)	52.7 ± 5.6 (17)	53.3 ± 8.0 (14)	47.0 ± 11.5 (19)
	12	Measurement	45.0 ± 10.3 (20)	43.3 ± 13.9 (18)	53.6 ± 6.9 (18)	51.2 ± 10.5 (13)	49.8 ± 8.0 (17)
		change from baseline/ *p*-value vs. placebo	−2.5 ± 13.0	2.2 ± 11.9/0.25	0.7 ± 6.6/0.36	−2.3 ± 10.6/0.96	1.3 ± 9.0/0.31
role-emotional	0	Measurement	45.8 ± 13.4 (20)	40.4 ± 13.9 (17)	47.8 ± 12.1 (17)	45.4 ± 16.2 (14)	44.3 ± 13.5 (19)
	12	Measurement	46.2 ± 10.1 (20)	45.4 ± 13.2 (17)	52.1 ± 9.1 (18)	43.9 ± 14.1 (13)	44.1 ± 14.0 (17)
		change from baseline/ *p*-value vs. placebo	0.4 ± 9.0	4.9 ± 12.8/0.22	3.9 ± 8.9/0.23	−5.3 ± 18.3/0.25	−1.5 ± 17.4/0.67
mental health	0	Measurement	52.4 ± 9.2 (20)	50.6 ± 10.4 (18)	55.5 ± 7.0 (17)	54.4 ± 10.4 (14)	53.4 ± 7.9 (18)
	12	Measurement	52.9 ± 7.5 (20)	52.6 ± 8.9 (18)	57.4 ± 6.9 (18)	54.3 ± 11.6 (12)	52.9 ± 8.6 (17)
		change from baseline/ *p*-value vs. placebo	0.5 ± 6.0	2.0 ± 11.1/0.61	1.8 ± 7.1/0.55	−0.5 ± 14.2/0.79	−0.3 ± 9.4/0.74
BDI (total score)	0	Measurement	9.5 ± 5.9 (19)	12.7 ± 8.4 (17)	7.8 ± 7.0 (16)	6.4 ± 6.4 (14)	8.9 ± 7.5 (18)
	12	Measurement	8.8 ± 6.9 (18)	10.9 ± 6.6 (17)	5.9 ± 4.2 (17)	5.8 ± 6.2 (13)	8.3 ± 8.2 (16)
		change from baseline/ *p*-value vs. placebo	−1.3 ± 4.5	−0.9 ± 6.6/0.86	−1.3 ± 5.1/0.97	1.1 ± 4.3/0.16	0.1 ± 3.0/0.31
EQ-5D-5L							
index value	0	Measurement	0.831 ± 0.138 (20)	0.784 ± 0.198 (18)	0.851 ± 0.199 (17)	0.853 ± 0.116 (14)	0.841 ± 0.185 (19)
	12	Measurement	0.851 ± 0.141 (20)	0.827 ± 0.155 (18)	0.889 ± 0.135 (18)	0.903 ± 0.098 (13)	0.837 ± 0.168 (17)
		change from baseline/ *p*-value vs. placebo	0.019 ± 0.125	0.043 ± 0.119/0.56	0.041 ± 0.167/0.65	0.023 ± 0.095/0.94	−0.034 ± 0.082/0.14
VAS	0	Measurement	73.2 ± 12.8 (20)	71.9 ± 12.4 (18)	72.7 ± 19.9 (15)	74.6 ± 17.9 (14)	73.7 ± 17.5 (19)
	12	Measurement	73.4 ± 11.1 (19)	73.3 ± 13.8 (18)	81.9 ± 12.6 (18)	80.4 ± 13.1 (13)	78.8 ± 17.4 (16)
		change from baseline/ *p*-value vs. placebo	0.5 ± 9.3	1.4 ± 15.5/0.84	8.7 ± 11.7/0.030	6.7 ± 13.5/0.14	3.1 ± 10.0/0.44

Data are presented as mean ± standard deviation (n) for measurements and mean ± standard deviation for change from baseline. A two-sample *t*-test was used for intergroup comparisons with the placebo group. ALA, 5-aminolevulinic acid; SFC, sodium ferrous citrate; SF-36, MOS 36-item short-form health survey; NBS, norm-based scoring; BDI, Beck Depression Inventory; EQ-5D-5L, EuroQol 5 Dimension-5 Level; VAS, visual analog scale.

**Table 5 nutrients-15-02866-t005:** Adverse events.

	Placebo Group	ALA 50/SFC 29 Group	ALA 100/SFC 29 Group	ALA 150/SFC 29 Group	ALA 100/SFC 57 Group
Number of subjects in the safety analysis set	20	18	18	15	19
Death	0 (0.0)	0 (0.0)	0 (0.0)	0 (0.0)	0 (0.0)
Any adverse event	5 (25.0)	2 (11.1)	2 (11.1)	2 (13.3)	5 (26.3)
Any serious adverse event	0 (0.0)	0 (0.0)	0 (0.0)	1 (6.7)	1 (5.3)
Dizziness	1 (5.0)/0 (0.0)	1 (5.6)/0 (0.0)	0 (0.0)/0 (0.0)	0 (0.0)/0 (0.0)	0 (0.0)/0 (0.0)
Complication of vaccination	0 (0.0)/0 (0.0)	1 (5.6)/0 (0.0)	0 (0.0)/0 (0.0)	0 (0.0)/0 (0.0)	1 (5.3)/0 (0.0)
Gastric disorder	0 (0.0)/0 (0.0)	0 (0.0)/0 (0.0)	0 (0.0)/0 (0.0)	0 (0.0)/0 (0.0)	1 (5.3)/0 (0.0)
Diarrhea	0 (0.0)/0 (0.0)	0 (0.0)/0 (0.0)	0 (0.0)/0 (0.0)	1 (6.7)/0 (0.0)	0 (0.0)/0 (0.0)
Elevated blood zinc	2 (10.0)/0 (0.0)	0 (0.0)/0 (0.0)	0 (0.0)/0 (0.0)	0 (0.0)/0 (0.0)	0 (0.0)/0 (0.0)
Bone fracture at shoulder	0 (0.0)/0 (0.0)	0 (0.0)/0 (0.0)	0 (0.0)/0 (0.0)	1 (6.7)/1 (6.7)	0 (0.0)/0 (0.0)
Hypertension	0 (0.0)/0 (0.0)	0 (0.0)/0 (0.0)	0 (0.0)/0 (0.0)	0 (0.0)/0 (0.0)	1 (5.3)/0 (0.0)
Upper respiratory infection	0 (0.0)/0 (0.0)	0 (0.0)/0 (0.0)	0 (0.0)/0 (0.0)	0 (0.0)/0 (0.0)	1 (5.3)/0 (0.0)
Hypoglycemia	1 (5.0)/0 (0.0)	0 (0.0)/0 (0.0)	0 (0.0)/0 (0.0)	0 (0.0)/0 (0.0)	1 (5.3)/1 (5.3)
Soft stool	1 (5.0)/0 (0.0)	0 (0.0)/0 (0.0)	0 (0.0)/0 (0.0)	0 (0.0)/0 (0.0)	0 (0.0)/0 (0.0)
Elevated white blood cell	0 (0.0)/0 (0.0)	0 (0.0)/0 (0.0)	1 (5.6)/0 (0.0)	0 (0.0)/0 (0.0)	0 (0.0)/0 (0.0)
Anemia	0 (0.0)/0 (0.0)	0 (0.0)/0 (0.0)	1 (5.6)/0 (0.0)	0 (0.0)/0 (0.0)	0 (0.0)/0 (0.0)
Constipation	1 (5.0)/0 (0.0)	0 (0.0)/0 (0.0)	1 (5.6)/0 (0.0)	0 (0.0)/0 (0.0)	0 (0.0)/0 (0.0)

Data are presented as the number of participants (%) who experienced non-serious or serious adverse events. ALA, 5-aminolevulinic acid; SFC, sodium ferrous citrate.

## Data Availability

The datasets generated or analyzed during this study are not publicly available owing to the absence of a statement in the informed consent documents and in the study protocol enabling data sharing with a third party after the end of the study, as well as the absence of approval for data sharing by the Juntendo University Certified Review Board.

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
