# Peer review of "Efficacy and Safety of 5-Aminolevulinic Acid Combined with Iron on Skeletal Muscle Mass Index and Physical Performance of Patients with Sarcopenia: A Multicenter, Double-Blinded, Randomized-Controlled Trial (ALADDIN Study)"

_nutrients, 2023, doi:10.3390/nu15132866_

Round 1

Reviewer 1 Report

Briefly, the article in question deals with a clinical trial that evaluated the effect of administering 5-aminolevulinic acid combined with ferrous sulfate on muscle mass index and handgrip strength in patients with sarcopenia. It is well reasoned and well written, in addition to providing important information on the subject. Therefore, I consider it suitable for publication. Here are some considerations:

 Title: adequate.

Abstract: adequate.

Introduction:

• In general, it presents coherent considerations that support the work;

Lines 53, 54 and 55: I understand that the definition of sarcopenia is referenced, but since mass loss does not always initially result in functional loss, the definition could be rethought and

Lines 80 and 81: Include that the objective was to evaluate the efficacy and dose response of ALA combined with Iron in patients with sarcopenia.

Material and methods:

Item 2.2: Inclusion criteria are well described but it was not clear whether patients should meet all criteria or just some of them. Were these criteria applied in sequence? Was there an algorithm for applying muscle function tests and BIA? This could be described;

Item 2.3: Randomization/allocation of participants into groups is a fundamental step in conducting clinical trials and is well described in this item. However, as a suggestion, wouldn't it be interesting to describe how the participants received the treatment/placebo? Was it by envelope or container? Did treatment and placebo have the same visual characteristics?

Item 2.4: Outcomes are well described. However, the primary outcome was assessed by BIO, which may not be reliable in some cases, so secondary outcomes are important. In this sense, it was not clear if only the BIO was evaluated before the intervention or if all outcomes were evaluated pre and post intervention. Considering Table 4, it seems to me that they were all evaluated, but this could be more explicit in the text.

Item 2.5: Was the analysis done by intention to treat? It seemed so to me but this needs to be highlighted as it is important for a clinical trial. Even if the statistical analysis was performed by a third-party entity, it would be interesting to describe whether this entity had access to group identification or not (triple blinding).

Results:

Well presented. The tables are a little long due to the presentation of the many results of the study and, therefore, it would be difficult to shorten them.

Discussion:

Adequate, including study limitations, especially regarding the use of BIA.

Conclusions: appropriate and consistent with the results presented.

References: Would it be possible to add more references? If possible, more current references would be welcome.

Reviewer 2 Report

In the study I had the pleasure to review, the authors described a study aimed to improve patient's quality of life suffering from sarcopenia. It is one of the Geriatric Giants and a very underestimated problem. Currently, many older adults are frail or sarcopenic, so finding ways to support kinesiotherapy in increasing muscle mass is valuable.  Although the results of the study should be supported by a larger group of participants in the future, the study is well-designed and described and in agreement with current legal regulations. Moreover, the authors respect older patients calling them "older individuals" and avoid the pejorative term "elderly". 

Some typographic mistakes should be corrected, for instance there is  a connective "and" at the end of the authors list and no author after it.  Moreover the quality of Figure 2 should be improved. 

I suggest the acceptance of the present manuscript after minor revision.

Reviewer 3 Report

This is an interesting paper about the efficacy and safety of 5-aminolevulinic acid combined with 2 iron on skeletal muscle mass index and physical performance 3 of patients with sarcopenia.

This study aims to assess the efficacy and dose dependency of ALA in sarcopenia by measuring skeletal muscle mass index. The study is very interesting and the limitations of the paper are well described.

 I just suggest to implement the introduction and the discussion about the role of muscle strenght in older adults (10.1016/j.archger.2020.104109) and the impact of drugs (10.1007/s40266-018-0521-y) especially in case of multimorbidity (10.1007/s12603-010-0036-7).
